# Commercial Poultry Production Stocking Density Influence on Bird Health and Performance Indicators

**DOI:** 10.3390/ani10081253

**Published:** 2020-07-23

**Authors:** Stéphane Bergeron, Emmanuelle Pouliot, Maurice Doyon

**Affiliations:** 1Egg Industry Economic Research Chair, Université Laval, Québec, QC G1V 0A6, Canada; 2Department of Agricultural Economics and Consumer Science, Université Laval, Québec, QC G1V 0A6, Canada; emmanuelle.pouliot.1@ulaval.ca (E.P.); maurice.doyon@eac.ulaval.ca (M.D.)

**Keywords:** poultry, stocking density, averaged daily gain, commercial production, animal welfare, economic performance

## Abstract

**Simple Summary:**

Poultry production aims for a stocking density of birds that will promote bird welfare and remain economically efficient. Most research that has examined the link between stocking density and bird welfare has been conducted in experimental facilities with a much smaller area and flock size, compared to commercial facilities. The current study is based on a large set of data from commercial broiler facilities. The results suggest that the stocking density has little impact on the number of mortalities and the quality of the meat. However, we did find increased growth rates at higher stocking densities which contrast with previous published results.

**Abstract:**

This study examines poultry production stocking density (SD) effect on bird welfare and economic performance. It is based on a large dataset from commercial production including observations for 2.2 million male broilers and 2.3 million female broilers from 37 production sites, with SD ranging from 20.63 kg/m^2^ to 41.15 kg/m^2^. The data collection was originally motivated by a processor’s economic concerns that increasing SD could cause slower broiler growth, higher condemnations, and lower grade meat. The data was examined using several linear regressions to determine how production parameters impacted these performance indicators. Results regarding foot pad lesion, condemnations, and mortality rates are consistent with those found in the literature. However, we find that daily weight gain is positively associated with SD, contrasting with past experimental results. The difference between the scope of commercial and experimental productions is discussed as a possible reason for these conflicting results.

## 1. Introduction

To address concerns over animal welfare in broiler production, countries and associations have established maximum stocking densities (SD). In 2007, the European Union regulated on acceptable SD ranges (Council Directive 2007/43/EC) with a base SD set at 33 kg/m^2^. This limit can be increased up to 39 kg/m^2^ if mortality is kept below a certain threshold and production climatic parameters are monitored. A further increase in SD between 39 kg/m^2^ to 42 kg/m^2^ is possible if monitoring authorities can confirm low mortality rates and good management practices. According to the current Canadian guidelines [1], SD should not be above 31 kg/m^2^ unless environmental conditions (humidity, temperature) are monitored and controlled daily, and that access to water is recorded. In such conditions, target SD can be increased to 38 kg/m^2^.

Results in the literature not only indicate that SD may adversely impact bird welfare, but may also reduce average daily growth (ADG), implying the loss of economic efficiency. Previous research conducted in experimental settings found a quasi-linear decrease in ADG as SD increased [2,3,4] while other research found no association between SD and ADG [5,6]. Observations from a large dataset from commercial productions found a decrease in ADG only when SD was above 38 kg/m^2^ [7].

The current study was motivated by a processor’s concerns for economic performance at higher SD, since it was believed that this may cause slower bird growth, higher condemnation and reduced meat quality. This study is based on a large dataset from commercial broiler productions, which contributes to the literature by complementing existing results that are mostly from experimental facilities that are limited in space and flock size. While experimental settings facilitate the monitoring of the health of the flock and the control of environmental parameters, they have limitations in their ability to replicate the living conditions under commercial productions. Specifically, experimental pen sizes tend to be rather small. For example, Dozier et al. [3] used 5.57 m^2^ pens for their experimental production, Guardia et al. [5] 2.75 m^2^ pens, and Feddes et al. [2] 14 m^2^ pens. In contrast, Dawkins et al. [7] provide a large-scale study with the cooperation of ten major commercial producers, including data from 2.7 million birds raised at production sites, with surface areas ranging from 455 m^2^ to 1901 m^2^. These differences in surface area could be of importance in the observed results, given that recent research on spatial occupation of broilers indicate that the total area, as well as the complexity of the spatial layout, influences birds’ behaviors and how they occupy the space [8]. To illustrate, Giersberb et al. [8] found that at a SD of 39 kg/m^2^, the area of floor space that is free (i.e., not used by the birds) ranges between 33.8 percent and 27.8 percent when the birds are respectively squatting or standing. Consequently, the SD between an experimental and commercial production could be the same, however, the total free surface area available to the birds is significantly different, and can play on the dynamics of the birds behaviors, disturbances, and movements [8]. In fact, it has been shown that the complexity of the production site layout influences location of crowding, with higher densities found next to heaters [9] and near wall enclosures [10]. Such complexities are usually not present in experimental pens that are of rectangular shape [2]. Therefore, these differences between experimental and farm settings may impact the measured effects of SD on bird health and carcass quality. In addition, a recent meta-analysis [11] found that the group size of the flock had significant impact on several performance indicators of bird health when interacted with SD. Experimental facilities have a median flock size of 60 birds per pen [11], compared to flock sizes ranging from 3410 birds to over 33,000 birds for the commercial productions used in the current study.

Our paper is based on a large dataset consisting of observations for 2.2 million male broilers and 2.3 million female broilers from 37 production sites, over a full calendar year. The data was recorded as part of a joint project between 37 broiler producers and a Canadian processor managed by a producer owned cooperative. The dataset has a SD ranging from 20.63 kg/m^2^ to 41.15 kg/m^2^, and the variables considered are: ADG, percentage of condemnations at the slaughterhouse, quality of meat (grade), foot pad lesions rates and mortality rates. Most results are consistent with the literature regarding foot pad lesion and mortality rates. However, our finding that daily weight gain is positively associated with increased SD, within the range tested, contrast with past experimental results found in the literature.

The paper is organized as follows. We first provide an overview of the data collection and present descriptive statistics of our sample. Then, a series of linear regressions of production characteristics on ADG, condemnations, quality of meat, pad lesions, and mortality rate are provided. The results are then contrasted with results reported in the literature. Finally, we discuss the implications of our results and offer concluding remarks.

## 2. Materials and Methods

The dataset is a collaborative work between a processor managed by a producer owned cooperative and participating member producers in Quebec, Canada. Data collection occurred between April 2018, and March 2019, with participating producers receiving, at each growing cycle, either male or female chicks of the same strain of Ross-308 bird allowing to treat SD effects according to sex, as suggested in the literature [4].

The dataset consists of independent observations measured at the slaughterhouse for each production facility per production cycle. Specifically, this corresponds to 197 series of observations from 37 production facilities, for a total of over 5.5 million birds, 2.2 million male broilers, and 2.3 million female broilers. While some producers have several production facilities, up to four, the data was entered for each facility independently given that the SD are specific to each facility. The total floor area of the production facility was provided by farm managers. The farm cooperative had the number of birds initially received (day 1) at the site, and the age of the birds at delivery. The remainder of the variables were recorded by employees along the processing chain at a single slaughterhouse. The employees of the slaughterhouse have specific task along the processing chain. The procedures used to obtain the data points in our model, such as classifying meat and removing condemned parts, are common task executed daily and under the supervision of veterinarians by federal law. Therefore, it can be assumed that the measurements are reliable.

The variables used were calculated as follows: The total weight of the flock was measured at the slaughterhouse reception and divided by the number of birds (includes mortalities at reception) to provide the end average weight. The mortality was calculated from the birds received at reception from the initial amount ordered. SD was calculated from the final weight of the flock divided by the producing facility floor surface area. ADG was calculated from the average weight at the slaughterhouse divided by the birds age at delivery (g/day). All birds are inspected by veterinarians after eviscerations. It is at this step that they may condemn birds, or parts of bird with an employee removing condemned parts. The condemned birds and parts were weighted. The total weight of condemnations divided by the weight of the eviscerated flock provided the percentage of condemnations. The veterinarian also assessed the presence of foot pad lesion, this is systematically done and is not specific to this study. It is also common practice to separate the Grade A meat from other meat along the processing chain. The weight of meat that did not reach Grade A status was used to calculate the percentage of Grade A meat.

The geographic area of the study has temperatures ranging from below −20 degrees Celsius in the winter to above 30 degrees Celsius in the summer. These extreme temperatures may pose difficulties in controlling the environmental parameters of the production. Therefore, we include the seasonality as a categorical variable in our model, defined according to which month the production cycle ended, with the first trimester defined as including April, May, and June; the second trimester including July, August, and September; the third trimester including October, November, and December; and the fourth trimester including January, February, and March.

Descriptive statistics with the sample minimums, maximums, medians, means, and variance of each variable are presented in Table 1. The average production site area is 1689 m^2^, and ranges from 255 m^2^ to 2604 m^2^. Male broilers were slaughtered at an average age of 35 days and female broilers at an average age of 38 days, at an average weight of 2.35 kg and 2.41 kg, respectively. SD varied from 20.63 kg/m^2^ to 41.15 kg/m^2^ (the maximum SD in Canada is 38 kg/m^2^, however some SD in our sample are higher than this limit and are caused by a delay in slaughtering), with an average of 32 kg/m^2^ for male broilers and 33 kg/m^2^ for female broilers. It is important to note that producers determined their own SD, according to their self-assessed ability at properly controlling environmental variables. This differs from other reported commercial results in the literature that imposed the level of SD as part of the research protocol. Figure 1 illustrates the distribution of SD against the production sites, revealing that, for smaller sites (<2000 m^2^), the SD varies from 25 kg/m^2^ to 40 kg/m^2^, while for larger sites (>2000 m^2^), the SD is mostly below 35 kg/m^2^. For this reason, a variable on production site area is included in the model.

The data was analyzed using five linear regressions (the relationship between the explanatory variables and the SD may not fit a linear relationship, however, since our analysis is restricted to SD between 25 kg/m^2^ and 40 kg/m^2^, we can assume that most relationships, if they exist, will be locally monotonic and a linear approximation will inform us on the sign of the association), using the R statistical package, one for each of the following dependent variable: ADG, condemnations, grade A, Pad-0 and mortality rate, each with the following explanatory variables: age, area (m^2^), SD (kg/m^2^) and seasonality, as defined above. A standard assumption about the normal distribution of residuals was made, since no collinearity between variables was found. An examination of a scatterplot of residuals versus the predicted values did not suggest that we should reject the assumption of homoscedasticity. We report the coefficients and the model F-Statistic and its associated *p*-value. The full statistics of each model including standard errors and t-values are available in the Appendix A
Appendix A.

## 3. Results

The coefficients of the five linear regressions are reported in Table 2 for both male and female models treated independently, with the column headings indicating the dependent variable.

The ADG regression models suggest that the coefficients for SD are statistically significant (*p*-value < 0.001), with values of 0.54 in the male model and 0.39 in the female model, suggesting that increasing SD, within the limits of this study, accelerates average growth rates. Furthermore, the third trimester (i.e., fall) shows the fastest growth with the coefficient being statistically significant (*p*-value < 0.05) in the female model. These results contrast with previous published studies. Guardia et al. [5] found no effect of SD on ADG, while most other studies found a negative correlation. For instance, Feddes et al. [2] found an average decline of 2.5 percent ADG with each 6 kg/m^2^ increase in SD. Similarly, Dozier et al. [3] found an average of 2 percent decline for every 5 kg/m^2^ increase in SD, while Zuowei et al. [4] measured a decrease of 5.5 percent ADG between their two extreme SDs of 26 kg/m^2^ and 42 kg/m^2^. The only other commercial production study [7] found that increasing SD does not impact ADG when below 38 kg/m^2^, but reports a 2 percent decrease in ADG from 38 kg/m^2^ to 42 kg/m^2^. Table 3 presents a summary of results from the literature related to the effect of SD on selected variables for the production of the Ross species.

The condemnations model indicates that SD had no effect on the percent of condemnations for male and female broilers. The only significant effect is with regards to the female broilers having higher condemnations when delivered to the slaughterhouse in the fourth trimester (i.e., winter), compared to the first trimester, which is the baseline. Note that winter transportation might have some impact on this result. The lack of links between condemnations and SD is consistent with Feddes et al. [2], the only other study found to have reported condemnation data.

The regression model on the percent of ‘Grade A’ meat was not influenced by any of the explanatory variables in both the male and female models, as indicated by Table 2. Again, this is consistent with previous results [4,6].

The regression model with regards to zero foot pad lesions as dependent variable suggest a seasonality effect, with percent of zero foot pad lesion decreasing in the second trimester (summer) for both male and female broilers, with both models finding a statistically significant negative coefficient (*p*-value < 0.05). Note that a negative coefficient here corresponds to increased observation of foot pad lesions, since our variable is the percent of zero foot pad lesions. No correlation was found with percentage of zero foot pad lesions and SD for the measured ranges of SD. This is consistent with results from Dawkins et al. [4], who found increased pad lesion at SD density only above 42 kg/m^2^, which is the upper limit of our sample. Jones et al. [12] also found that foot pad lesions increased in summer months. In contrast, Dozier et al. [3] found that foot pad lesion increased linearly with SD. However, they indicated that environmental factors other than SD were not controlled and indicated higher litter moisture at higher SD could possibly explain the observed increase in foot pad lesions.

## 4. Discussion

The data used for this analysis differs from most published papers that report results from experimental productions conducted at much smaller scales. Previous studies using data from commercial productions imposed a SD on producers [4,12], while the current study left the SD choice up to producers. This difference can have a significant impact on observed results, since producers are self-aware in their ability at controlling environmental parameters, and are best suited at determining which SD they could properly manage. As Dawkins et al. (2004) indicate, the environmental growth conditions may play a larger role in bird welfare than SD. This is not to say that SD is unimportant, as the control of humidity, air quality, and temperature is increasingly complex at higher SD. However, our results suggest that SD, within the range studied, has no or little impact on mortality rates, meat quality, and the percentage of zero foot pad lesion.

Most results in the literature, based on experimental facilities, find a negative correlation between SD and ADG [2,3,4]. However, results from a commercial production [7] found a decrease in ADG only at SD above 38 kg/m^2^, and none otherwise. Our result contrast with these past results, finding a positive ADG with increasing SD. The fact that this association is found for both the male and female datasets offers a replication that strengthens the confidence of the observed finding.

The difference between commercial and experimental facilities may explain these differences. Based on planimetric analysis [8], the area of floor space that is free (i.e., not used by the birds) at an SD of 39 kg/m^2^ ranges between 40.4 percent of total area when the birds are standing to 36.3 percent when they are squatting. At an SD of 33 kg/m^2^, the free floor space corresponds to 51.5 percent and 48.2 percent of total floor space area for standing and squatting birds, respectively (these percentages correspond to the measured free space when target end weights are 2500 g, as reported in Table 3 of Giersberg et al. [8]). Using these values to calculate available floor space at the end of cycles, we found that, for the largest production site in our sample (2604 m^2^), an SD of 39 kg/m^2^ provides 1052 m^2^ of free floor space when the birds are standing, and 945 m^2^ when they are squatting. In comparison, we calculated the area of free space of the average production sites in our sample (1629 m^2^) at an SD of 33 kg/m^2^ yields 839 m^2^ of free floor space when standing and 591 m^2^ of free floor space when squatting. These calculations demonstrate that larger facilities provide more total free space at higher densities, compared to smaller facilities with lower densities. Moreover, the consideration of production site total area and the complexities of layout design have been shown to influence birds’ behaviors, location of crowding and occurrences of disturbances [9,10]. Therefore, studies that control for SD in limited experimental production facilities inadvertently influence variables, such as total area of free space, complexity of the layout, and flock size, which have all been shown to influence bird health and performance [11].

This study was motivated by economic concerns that increased SD adversely affected broiler growth at the farm and the efficiency at the slaughterhouse through higher condemnations and lower grade meat. The results from our analysis suggest that higher SD, within the range studied, has little or no effect on these variables and should not reduce efficiency at the slaughterhouse. The higher ADG observed might suggest potential economic gains at higher SD with little impact at the slaughterhouse. However, caution is needed when interpreting these results. It is important to note that these results are conditional on farmer’s ability to control environmental conditions that maintain bird health at these higher densities. The data used reflects the fact that the participating sites at higher SD were already operating at these levels of SD, with the managerial skill and equipment necessary to do so. While the current study did not calculate impacts of higher SD on cost of production, and more specifically feed conversion ratios (FCR), a review of results suggest that FCR remains relatively constant when SD is increased if environmental controls are used [11]. Again, these last findings are mostly based on experimental production and would need to be validated with observations obtained from commercial facilities.

The current analysis has its limitations. Several observations are from multiple production cycles of the same producer and at the same production site. Therefore, if more data were available, more sophisticated hierarchical models could be used to identify site-specific effects. Furthermore, the ability to identify seasonal effects is limited since the dataset only spans one year and might be influenced by particularities of the measured season. A study spanning several years would be needed.

## 5. Conclusions

The large datasets of observations from commercial facilities complement existing results from the literature that are mostly obtained from experimental units of smaller scope. While experimental settings with controlled pens are essential to advance the understanding of environmental effects on bird health, complementary observations from commercial facilities provide a validation of these results in environments where bird behavior may be different. Future studies at commercial scales would be needed, with a more complete set of explanatory variables, in order to better understand how environmental characteristics, including humidity and house layout, interact with SD and the implications these interactions have for bird welfare and economic performance.

## Figures and Tables

**Figure 1 animals-10-01253-f001:**
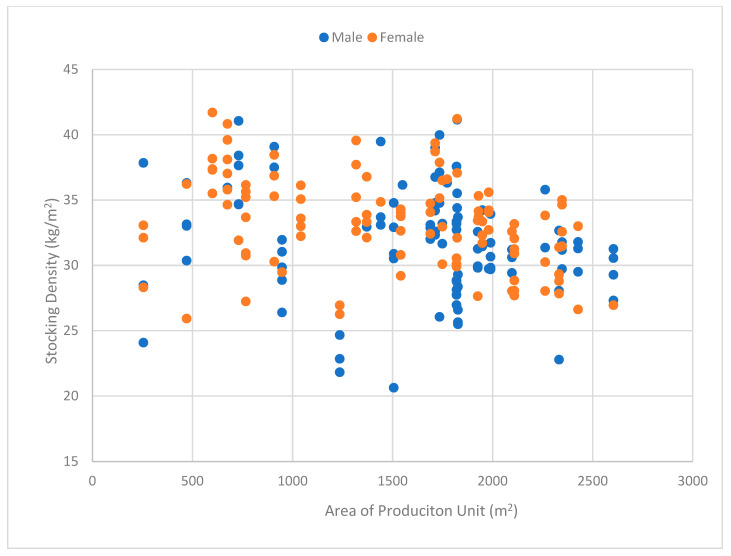
Distribution of the observations, with stocking density (kg/m^2^) over the area (m^2^) of the production unit, color coded by gender.

**Table 1 animals-10-01253-t001:** Descriptive statistics (minimum, maximum, median, mean, and variance) of sample variables.

	**Male Broilers**				
	**Min**	**Max**	**Median**	**Mean**	**Variance**
Age (days)	32	39	35	35.4	2.17
End Weight (kg)	1.49	2.96	2.41	2.41	0.031
Area (m^2^)	255	2604	1773	1629	341,110
Stoking Density (kg/m^2^)	20.63	41.15	31.78	31.85	17.5
ADG (g/day)	43.70	80.04	67.60	67.86	2.00
Condemnations	0.50%	15%	1.60%	2%	2%
Grade A (%)	82.80%	98.70%	93.00%	93.90%	6.90%
Pad-0 (%)	0.00%	100.00%	68.00%	66.80%	7.70%
Mortality (%)	0.00%	13%	3.70%	4.30%	0.04%
	**Female Broilers**				
	**Min**	**Max**	**Median**	**Mean**	**Variance**
Age (days)	35	42	38	38	2.06
End Weight (kg)	1.89	2.72	2.35	2.35	0.026
Area (m^2^)	255	2604	1689	1508	392,402
Stocking Density (kg/m^2^)	25.9	41.7	33.3	33.2	12.5
ADG (g/day)	52.49	69.0	62.3	61.7	8.4
Condemnations (%)	0.30%	3.80%	1.20%	1.40%	0.0004%
Grade A (%)	85.20%	98.67%	93.00%	92.90%	7.74%
Pad-0 (%)	0.00%	100.00%	88.60%	76.90%	6.50%
Mortality (%)	0.20%	18.00%	2.90%	3.20%	0.05%

ADG: average daily gains; % Pad-0 is the percentage of zero foot pad lesions.

**Table 2 animals-10-01253-t002:** Coefficient of linear regression models with dependent variables name as column headers.

	Male Broilers Regression Models									
Variables	ADG		Condemnations		% Grade-A		% Pad-0		Mortality Rate	
Intercept	40.31	***	−0.34		84.68	***	56.87		−5.02	
Age	0.26		0.00		0.21		0.73		0.33	#
Area	0.00		0.00		0.00		0.00		0.00	
Density	0.54	***	0.07		0.04		-0.14		−0.11	#
Trimester										
2nd	0.76		0.03		2.00	*	−19.52	*	0.08	
3rd	2.09	#	0.68		0.83		−15.46		0.26	
4th	1.05		0.74		0.97		1.74		1.05	#
F-Statistic	8.42		1.46		1.20		2.02		1.55	
*p*-value	0.00	***	0.20		0.32		0.07		0.009032	**
	**Female broilers regression models**									
	ADG		Condemnations		% Grade-A		% Pad-0		Mortality Rate	
Intercept	39.26	***	0.38		101.30	***	152.69	*	−5.60	
Age	0.22		0.02		−0.10		−1.18		0.42	*
Area	0.00		0.00		0.00		−0.01		0.00	
Density	0.39	***	0.01		−0.12		−0.42		−0.22	**
Trimester										
2nd	−0.02		−0.29	#	0.11		−16.36	*	1.05	#
3rd	1.87	**	−0.27	#	0.75		−7.35		0.20	
4th	1.19	#	0.56	***	0.54		−2.79		0.68	
F-Statistic	10.56		8.473		0.6359		1.55		3.64	
*p*-value	4.88 × 10^−9^	***	1.991 × 10^−7^	***	0.7012		0.1698		0.00263	**

# *p* ≤ 0.10; * *p* ≤ 0.05; ** *p* ≤ 0.01; *** *p* ≤ 0.001.

**Table 3 animals-10-01253-t003:** Selected results from literature on the effect of stocking density on indicators of broiler health for the Ross species.

Reference	Stocking Density (kg/m^2^)	Age (days)	Area of House/Pen (m^2^)	Mortality	ADG	Pad Lesion
Dawkins et al. [7]	30, 34, 38, 42, 46	39–42	455–1901	No effect	Avg. decrease of 2% at SD > 38	Increases as SD > 42
Dozier et al. [3]	25, 30, 35, 40	36	5.57	Highest at SD 25	Avg. 2% decrease at each SD increase	Increase linearly with SD
Feddes et al. [2]	23, 29, 35, 46	37–39	14	No effect	Avg. decrease of 2.5% at tested SD > 29	NA
Goo et al. [6]	22, 26, 35, 41	21	0.59	NA	Inconclusive	NA
Guardia et al. [5]	31, 43	39	2.75	NA	No effect	NA
Zuowei et al. [4]	26, 42	42	3	NA	Avg. of 5.5% between the two SD.	NA

ADG: average daily gains; % Pad-0 is the percentage of zero foot pad lesions; NA: Not available.

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
