# Peer review of "Commercial Poultry Production Stocking Density Influence on Bird Health and Performance Indicators"

_animals, 2020, doi:10.3390/ani10081253_

Round 1
Reviewer 1 Report
This is an interesting paper that looks at the relationship between stocking density and various health and production parameters in commercial broiler chickens. This an important subject relevant to this journal but the paper contains a serious methodological flaw which means that the conclusions are not justified by the results.
The authors claim that, contrary to other studies, they did not find that growth rate declined at higher stocking densities and they therefore argue, on the basis of this single study, that farmers can keep birds at higher stocking densities. The problem is that the stocking densities were not allocated by the researchers but the participating farmers were left to decide their own stocking densities. This means that farmers who knew from past experience that they could safely keep birds at higher stocking densities without seeing a drop in growth rate would choose to do so but it does not follow, as the authors argue, that other farmers could raise their stocking densities and not see a decline in growth rate. In my view the paper should not be published in its present form because it claims more than is justified from the results. A single finding that goes contrary to other research results and has such a profound methodological problem should not advocate something as controversial (to bird welfare and farmer economics) as an increase in stocking density, as it could have potentially damaging consequences. I strongly recommend that the paper be re-written with the controversial advice to farmers being substantially modified.
Methods: The fact that the farmers themselves chose their own stocking densities should be made explicit in the Methods and not left to the Discussion.
Line 94: who did the data collection? Was it the researchers or did they rely on data routinely collected at the farm and the abattoir? Were all flocks slaughtered in the same place with the same measurements taken? More information should be given about how data was collected.
Author Response
Please see attachement.

Reviewer 2 Report
See attached marked up MS
Review of Manuscript “Commercial poultry production stocking density 2 influence on bird health and performance indicators”
I made corrections to the text of the PDF version on the following lines:
9, 11, 12, 13, 14, 49, 50, 53, 54, 63, 65, 69, 72, 73, 84, 85, 86, 117, 179, 187, 188, 192, 196, 197, 200, 213, 214, 221, 222, 224,
These are mostly English grammar issues to clarify the text.
I also do not think the values in Figure 1 (Table) for the Females is correct:
MIN MAX Median Mean
|
Condemnations (%) |
0.30 |
0.04 |
0.01 |
0.01 |
|
Grade A (%) |
0.85 |
0.99 |
0.93 |
0.93 |
|
Pad-0 (%) |
0.00 |
1.00 |
0.89 |
0.77 |
|
Mortality (%) |
0.00 |
0.18 |
0.03 |
0.03 |
These values seem to be decimal points off compared to the male data for the same. This is saying that the mortality for female broilers is only 0.18% when the males had 13% mortality?
Check these values again and make sure your decimal points are in the correct place.
Too bad you did not include FCR since you think there may be an economic gain to increasing the kg/m2

Reviewer 3 Report
Overview & General Recommendations
This manuscript describes the analysis of commercial broiler production field data to understand the effect of stocking density on broiler health, welfare, and performance. There is a large amount of commercial production field data that is often unexplored; therefore, this manuscript and the analyzed dataset is to some extent novel. There is growing interest in this area and these analyses can offer new insights and lead to improvements on broiler health, welfare, and performance. Because of this, the current study is on a topic of relevance and general interest to the readers of this journal.
The manuscript reads satisfactorily, with a few technical and non-technical English language and style errors. The manuscript is logically organized and flows clearly between sections. The content of each section is well-organized. References appear to have correct formatting and are complete.
There are some key considerations in the data preparation and quality that must be addressed.
Major comments
- For the selected sample variables, which were assessed per bird (i.e., resulting 5.5M observations) and which were measured at the “house level”? This should be clarified.
- Data Quality. Often with commercial production data, the individual site manager and/or processing plant workers are responsible for data entry which can be affected by training, level of experience, etc. This tends to lead to high variability. Can you provide insight to the variability inherent in this dataset? The descriptive statistics are provided in table 1, but there is no estimation of variance. Was there any missing data?
- Also, was there any statistical analysis to detect outliers? Was the dataset assumed to be normally distributed? How was this assessed? Were any transformations needed, such as converting percentages to proportion values? With this dataset, there are typically numerous “zero” values that require transformation.
Minor comments
Do you have information regarding the particular line of Ross broilers that were used, e.g., Ross 308, 708, etc.?
I was unable to access the supplementary materials.
If additional visualizations of the data and the analysis could be included, it would be well received.
Please add commas for numbers greater than 999.
Sporadically throughout, some values that should be superscripted (e.g., kg/m2) are not, please correct.
L14, remove strikeout
L35, no need to define SD twice
L41, condition should be plural
L76; L90, change ‘run’ to ‘operated’ or ‘managed’
L91, change ‘ran’ to ‘occurred’
L96, add ‘area’ after surface
L195, it is unclear how stocking density and housing floor area is related to the 11,460 m2 value that is estimated in this sentence.
table 1, change ‘density’ to SD or stocking density; values under male broiler have ‘%’ while values under female broilers do not
table 3, add ‘y’ to Densit; make 2 superscript in m2
table 5, missing ‘5’ for *P
Round 2
Reviewer 1 Report
The authors have substantially modified the text in the light of the criticism that they previously claimed too much from their results. They have acknowledged the limitations of their study and reduced the recommendations that should follow from them. The data presented are a very useful contribution to the growing body of evidence of the complexity of this issue of stocking density and welfare and I therefore recommend that it should now be published.
Author Response
no further comments
Reviewer 3 Report
The authors have improved the manuscript in responses to the reviews. Namely, the materials and methods section has key additional details and justification for the approaches selected. The discussion has been enhanced as well. This is an interesting study and a well-received contribution to this topic area. There are a few typos throughout.
Author Response
no further comments